# Human Orf with Immune-Mediated Reactions: A Systematic Review

**DOI:** 10.3390/microorganisms11051138

**Published:** 2023-04-27

**Authors:** Luca Rossi, Giorgio Tiecco, Marina Venturini, Francesco Castelli, Eugenia Quiros-Roldan

**Affiliations:** 1Department of Clinical and Experimental Sciences, Unit of Infectious and Tropical Diseases, University of Brescia and ASST Spedali Civili di Brescia, 25123 Brescia, Italy; l.rossi029@unibs.it (L.R.); g.tiecco@unibs.it (G.T.); francesco.castelli@unibs.it (F.C.); 2Department of Clinical and Experimental Sciences, Section of Dermatology, University of Brescia, 25123 Brescia, Italy; marina.venturini@unibs.it

**Keywords:** Orf, ecthyma contagiosum, pustular dermatitis, bullous, autoimmune, erythema multiforme, pemphigoid, blister, systematic review, review

## Abstract

*Background:* Orf is a highly contagious zoonosis caused by Orf virus (ORFV), which is endemic in sheep and goats worldwide. Human Orf is usually a self-limiting disease, but potential complications, including immune-mediated reactions, may occur. *Methods:* We included all articles regarding Orf-associated immunological complications published in peer-reviewed medical journals. We conducted a literature search of the United States National Library of Medicine, PubMed, MEDLINE, PubMed Central, PMC, and the Cochrane Controlled Trials. *Results:* A total of 16 articles and 44 patients were included, prevalently Caucasian (22, 95.7%) and female (22, 57.9%). The prevailing immunological reaction was erythema multiforme (26, 59.1%), followed by bullous pemphigoid (7, 15.9%). In most cases, the diagnosis was made on the basis of clinical and epidemiological history (29, 65.9%), while a biopsy of secondary lesions was performed in 15 patients (34.1%). A total of 12 (27.3%) patients received a local or systemic treatment for primary lesions. Surgical removal of primary lesion was described in two cases (4.5%). Orf-immune-mediated reactions were treated in 22 cases (50.0%), mostly with topical corticosteroids (12, 70.6%). Clinical improvement was reported for all cases. *Conclusions:* Orf-related immune reactions can have a varied clinical presentation, and it is important for clinicians to be aware of this in order to make a prompt diagnosis. The main highlight of our work is the presentation of complicated Orf from an infectious diseases specialist’s point of view. A better understanding of the disease and its complications is essential to achieve the correct management of cases.

## 1. Introduction

Orf is a highly contagious, zoonotic, self-limited skin infection caused by Orf virus (ORFV), which is an epitheliotropic DNA virus belonging to the Parapoxvirus genus of the Poxviridae family. In the literature, Orf is also known as “ecthyma contagiosum”, “infectious pustular dermatitis”, “contagious pustular dermatitis”, or “scabby mouth” (in sheep and goats) [1]. Current evidence suggests the existence of two different types of viruses infecting sheep (type S) and goats (type G), originated in the 1800s in different geographic areas (Europe and Asia, respectively) [2]. Currently, the infection is endemic in sheep and goats worldwide, mainly affecting young animals in the first year of their life with a high risk of outbreaks in settings of intensive sheep husbandry [3,4]. Occasional infections have been also reported in camels, Japanese serow, and cats [5]. Moreover, some cases of Orf have been described in Alpine chamois (*Rupicapra rupicapra*) in Middle Europe, with exceptional transmission to hunters [6].

In animals, the virus typically manifests as scabbed sores on or around the mouth, but may also present on the legs or udders. Despite it remaining localized in the epithelium and rarely leading to fatal outcomes, mortality can reach 90% in young animals, usually due to oral lesions that impede suckling, secondary bacterial and fungal infections, or maggot infestation [4].

Moreover, due to its resistance to harsh environmental conditions and several immune evasive mechanisms, it can repeatedly reinfect sheep and goats, contributing to the further spread of infections in other animals, such as deer, guinea pigs, dogs, and camels, as well as humans [7].

All of these characteristics make ORFV infection in herds a serious sociocultural and economic challenge for livestock farmers. In low-income communities, vaccination of the herds is highly recommended in the absence of a specific antiviral treatment, although current commercially licensed live-attenuated vaccines have partial efficacy, short-lived immunity, and return to virulence [8,9].

Transmission to humans can occur through inoculation of broken or abraded skin by direct contact with infected animals, meat, or carcasses, or indirectly with contaminated fomites. ORFV has been found to survive for up to 17 years in environments with a dry climate, remaining viable on the wool of animals and contaminated material for significant periods [8]. Autoinoculation and human-to-human transmission are rare, but are occasionally described [10]. Recently, Coradduzza et al. suggested that only specific viral strains, with high levels of virulence, could infect humans [2].

Orf is considered an occupational disease, and at-risk populations include mostly shepherds, wool shearers, butchers, farmers, and veterinarians. However, infection may also occur in individuals with nonoccupational contact with sheep and goats, such as zoologic garden visitors and people who slaughter animals for traditional rituals [4]. Approximately 3 to 7 days after inoculation, infection in humans usually presents as a single papule or nodule on the hands or fingers, evolving through six different stages until spontaneous resolution in approximately 6–8 weeks [1]. Multiple lesions may also occur. Moreover, persistent or more aggressive forms of disease, such as large highly vascularized tumor-like skin lesions, can exceptionally present in immunocompromised patients, especially those with T cell dysfunction [4]. Sometimes, it may be associated with local symptoms such as pain and pruritus, or less frequently with systemic symptoms such as fever or malaise [1,11].

Human Orf lesions can resemble other localized poxvirus infections, such as cowpox, pseudocowpox (milker’s nodule), and monkeypox, as well as more serious conditions including anthrax, tularemia, primary inoculation tuberculosis or atypical mycobacterial infections, ecthyma gangrenosum, syphilis, sporotrichosis, pyogenic granuloma, acute febrile neutrophilic dermatosis (sweet syndrome), and neoplasia [1,12,13]. Potential complications may occur, including secondary bacterial infections, lymphadenopathy, lymphangitis, and secondary immunological manifestations such as erythema multiforme (EM), widespread papulovesicular eruption, Stevens–Johnson syndrome, or antibody-mediated hypersensitivity reactions such as blistering disorders [1,14]. The most frequently reported immunological complication (7–18%) is EM, a self-limited acute condition with a wide spectrum of severity [4]. It is considered a type IV hypersensitivity reaction, which typically presents with targetoid lesions on the skin and sometimes on the mucosa, usually following exposure to certain infections (90% of cases), and less commonly to drugs [15,16]. The diagnosis in humans is usually based on the anamnesis and the clinical features [17]. However, because of the extensive clinical differential diagnosis of primary lesions and secondary immunological reactions, as well as the unfamiliarity of many physicians with this disease, especially in areas where it is uncommon, currently Orf is under-recognized, often leading to delayed diagnosis and inappropriate treatments [18,19].

Despite the hypersensitivity reactions associated with Orf resembling cell-mediated immune reactions seen in other viral infections, the pathophysiology is still unclear. Mechanisms of Orf-induced autoimmune diseases may include viral mimicry of host proteins (‘molecular mimicry’) or alteration of basement membrane proteins by the virus (increasing immunogenicity) [20]. Moreover, considering that the unaware clinician might prescribe drugs for an autoresolving primary lesion, the occasional immune-mediated skin rashes that may occur during Orf infections are under-reported in the literature as mistakable with drug allergic events [14,18,21,22].

All things considered, our primary aim is to systematically review the current literature to provide an update on clinical presentation, evolution, diagnosis, treatment, and outcome of Orf complicated by secondary immune-mediated reactions.

## 2. Methods

Our methods meet the Preferred Reporting Items for Systematic Reviews and Meta-Analysis (PRISMA) updated guideline for systematic review stated in 2020 [23].

### 2.1. Eligibility Criteria

We included all case reports, case series, retrospective, and prospective human studies regarding Orf-associated immunological complications published in peer-reviewed medical journals. We excluded articles regarding ecthyma contagiosum without immunological complications. Articles published in non-English languages, preprint or ahead of print analysis, preclinical studies, animal studies, letters to the editor, conference articles, commentaries, viewpoints, reviews, systematic reviews, and meta-analyses were also excluded.

### 2.2. Information Sources and Search Strategy

An electronic search was employed to find the published articles which reported Orf-associated immunological complications through the United States National Library of Medicine, PubMed (last accessed 8 January 2023), MEDLINE (last accessed 8 January 2023), PubMed Central, PMC (last accessed January 2023), and the Cochrane Controlled Trials (last accessed 8 January 2023). References for this review were identified by searching for the following terms in the Title/Abstract section: “Orf” OR “ecthyma contagiosum” OR “infectious pustular dermatitis” OR “contagious pustular dermatitis” OR “thistle disease” OR “scabby mouth” were combined with “bullosa” OR “bullous” OR “immunobullous” OR “immuno bullous” OR “immuno-bullous” OR “autoimmune” OR “multiforme” OR “erythema multiforme” OR “pemphigoid” OR “blister” OR “blisters” OR “blistering” OR “erythematous” OR “erythematosus”.

### 2.3. Selection and Data Collection Process

A team of 2 resident doctors in Infectious and Tropical Diseases of the University of Brescia, Italy, read the abstract of each scientific work and independently selected the articles according to the established criteria (LR, GT). A Professor in Infectious and Tropical Diseases of the ASST Spedali Civili di Brescia, Italy (EQR), revised the included and the rejected papers. Then, resident doctors (LR, GT) considered the full text of each selected article to collect data that were revised, compared, and synthesized using a detailed database. Disagreement was resolved by a joint discussion that included all authors.

### 2.4. Data Items

For each selected article, we collected information regarding the number of patients with Orf disease, immune-mediated complications, and their demographic data (age, gender, and ethnicity). Regarding the transmission patterns, we reported information about animal sources and proven animal lesions. Relative to the clinical manifestations, we collected the characteristics of Orf and its immune-mediated reactions, specifying the time of onset from primary lesions. Furthermore, we assessed the diagnosis process by reporting histopathological and serological information, when available. Eventual treatments provided for primary lesions and for complications were considered. Finally, we reported the clinical outcome and the follow-up evaluations. Missing or unclear data were reported as “non-available” (NA).

### 2.5. Synthesis Methods

All the collected data were reported in a single table, where every column was specifically associated with a different item. We limited our study to a descriptive analysis of our search findings due to the wide heterogenicity of the selected articles. The percentage calculation was performed in consideration of the number of data available for each specific item. No models to identify the presence and extent of statistical heterogeneity, or sensitivity analyses to assess the robustness of the synthesized results, were performed.

### 2.6. Bias and Certainty Assessment

Since the wide heterogenicity of the selected articles, in this systematic review we have only performed a descriptive analysis of our findings. Risk of bias or certainty (or confidence) in the body of evidence was not assessed.

## 3. Results

### 3.1. Study Selection and Search Results

A total of 114 papers were identified through our search. Of these, 98 articles were excluded for several reasons, as shown in Figure 1. Thus, only 16 articles met the inclusion criteria and were considered in this review, as shown in the following flow-chart (Figure 1).

The included papers were eight case series (50.0%), seven case reports (43.8%), and one retrospective nonrandomized monocentric study (6.3%).

The summary of the characteristics and findings of the 16 included papers are presented in Table 1.

Most articles were published before 1993 (6/16, 37.5%) or after 2014 (7, 43.8%), almost all of them in dermatology journals (12, 75.0%), and only one (6.3%) in an infectious disease journal. Regarding the geographic distribution, most of the data comes from the Mediterranean area (5, 31.3%) and the Continental or Northern Europe (5, 31.3%). A minor part was written in the U.S.A. (3, 18.7%) and Iran (3, 18.7%).

### 3.2. Epidemiology

A total of 44 patients were included. Considering the available demographic data, the patients were prevalently Caucasian (22/23, 95.7%) and female (22/38, 57.9%). The mean age was 39.3 sd 15.7, with a median of 37 years old (range: 6 to 67). Most patients had no comorbidities (16/17, 94.1%). Only 20 patients reported a certain contact with animals, mainly sheep (55.0%) and lambs (25.0%), or with carcasses and raw meat (15.0%). Two patients (10%) also reported having contact with goats. Proven animal lesions were present in nine cases (45.0%).

### 3.3. Clinical Manifestations

Considering the 44 patients with documented Orf and immune complications, 29.5% presented a single primary lesion, while multiple lesions were described in 18.2% of cases. An unspecified number of lesions was reported in 23 patients (52.3%). Secondary immune-mediated reactions appeared more frequently within 3 weeks from the primary lesions (13, 68.4%), always involving the skin, especially the limbs and extremities (18, 40.9%). Eruption on the neck, face, or scalp occurred less frequently (12, 27.3%), while a generalized skin rash was rarely described (3, 6.8%). Mucosal involvement was reported only in six (13.6%) cases (Table 2).

Mostly, the secondary immune-mediated lesions were papules, macules, target lesions, blisters, or bullae. Often, two or more of these lesions were simultaneously present in the same patient. They were generally asymptomatic, with itching being reported in nine cases (20.5%). Regarding other Orf non-immune-mediated complications, four patients (9.1%) presented lymphangitis and lymphadenopathy, and local bacterial infection occurred in two cases (4.5%).

### 3.4. Diagnostic Approach

As shown in Table 3, the prevailing diagnosis was erythema multiforme, which occurred in 26 of 44 cases (59.1%), followed by bullous pemphigoid (7, 15.9%).

In most cases, the diagnosis was made on the basis of clinical and epidemiological history (29, 65.9%), while a biopsy of secondary lesions was performed in 15 patients (34.1%). In two cases, molecular tests were also used to support the diagnosis. As shown in Table 4, the main histopathological findings were eosinophilic/neutrophilic infiltrate and subepidermal blisters. In addition, in 10 patients, direct immunofluorescence on bioptic samples was performed, revealing deposition of IgG, C3, and/or IgA along the basement membrane zone. In one patient (2.3%) with immunobullous disease, antilaminin 332 circulating autoantibodies were searched for and found to be positive.

### 3.5. Treatment and Outcome

Regarding the therapeutic approach, 12 (27.3%) patients received an initial treatment for primary lesions. Most cases (10, 83.3%) were treated just with topical drugs, mainly using local antibiotics (5, 50.0%), imiquimod cream (3, 30.0%), or compresses of KMnO_4_ solution and alcohol iodine (3, 30.0%). One patient (8.3%) received silver nitrate. Less frequently, systemic treatments were administered in seven cases (58.3%), of which six (85.7%) received an oral antibiotic, and one (14.3%) received dapsone. Surgical removal of the primary lesion was described in two cases (4.5%).

Twenty-two patients (50.0%) received a pharmacological treatment. Most of them received a topical treatment (17, 77.3%), mainly including corticosteroids (12, 70.6%). In 12 cases, systemic treatment was provided, primarily using corticosteroids (75%) and/or antihistamine (50%), while methotrexate was administered in one patient (8.3%).

Finally, clinical improvement was reported for all cases, but data regarding resolution time and follow-up were available for only 14 patients (31.8%). In 10 cases (71.4%), resolution of the immune-mediated lesions occurred within 2 weeks, while it took longer for the remaining 4 patients (28.6%). The three patients having received topical imiquimod on primary lesions showed a clearance of both Orf and immune-mediated reactions slightly earlier (within 1–2 weeks). A follow-up visit within 3 months was made in nine patients (20.5%), showing complete healing and absence of relapses. Regarding the patient with antilaminin antibodies, two months after being treated with oral prednisone, dapsone, and topical clobetasol propionate, no circulating autoantibodies were detectable.

## 4. Discussion

This systematic review collected available data regarding human Orf complicated by secondary immune-mediated reactions. We examined all of the cases reported over a period of almost fifty years, analyzing each for patient clinical presentation, diagnosis, treatment, and outcome.

There is ample literature on human Orf; however, few papers are present regarding Orf-related immune reactions. Our research only found 16 articles, mostly case reports or case series.

Despite its exact prevalence being unknown, human Orf is primarily considered an occupational hazard for people having direct contact with infected animals [19,38]. The disease in humans usually undergoes spontaneous healing, with occasional complications [1]. Due to the self-limited natural history and the common awareness amongst rural communities where it is most prevalent, infected people often make a self-diagnosis and choose not to consult a doctor [18]. On the other hand, many physicians in areas where Orf is less frequent are unfamiliar with it, and this can lead to misdiagnosis, especially when immunological manifestations are also present [18]. All things considered, the low number of included papers may result from several factors, perhaps the main of which is a low incidence of Orf-mediated immune reactions.

Among the included papers, the majority were published before 1993 (37.5%) or after 2014 (43.8%), perhaps suggesting an increasing interest in an old and re-emerging diseases [39]. Moreover, almost all of them were published in dermatology journals, and only one in an infectious disease journal, probably reflecting the higher number of cases usually seen by dermatologists, as well as a lack of interest from other medical specialties, including infectious diseases. This, in addition to the low number of reported cases, may favor under-recognition and misdiagnosis of human Orf, suggesting the need to spread current evidence regarding this disease and its potential complications.

In this review, almost all of the cases were associated with sheep, with only two patients also reporting contact with goats. This result appears to be in contrast with the literature, which suggests that Orf infection originates from goats equally often [2]. This review has limited basis for inference; however, this finding may reflect the suggested occurrence of two genome types of ORFV (S from sheep and G from goats), which originated in different geographic areas [2].

According to the current evidence, ORFV can be associated with a wide spectrum of hypersensitivity reactions in humans, ranging from erythema multiforme to autoimmune bullous diseases [4]. In our systematic review, erythema multiforme was the most frequently reported immunological complication, always appearing before the complete resolution of primary lesions, and mostly within 3 weeks after the Orf onset. These hypersensitivity reactions are not Orf exclusive, but have been associated with other infectious diseases. This is particularly true for erythema multiforme, of which herpes simplex virus is considered the leading cause, with a mechanism that has been extensively studied [15,40,41]. Peripheral blood mononuclear cells carry HSV-DNA fragments to distant skin sites where they are expressed on keratinocytes, leading to the recruitment of HSV-specific CD4+ Th1 cells [41,42]. The resulting T-cell-mediated response brings the formation of sub- and intraepithelial vesiculation, which clinically manifest as blistering and erosions [41]. It remains unclear how this virus can disseminate and then reach the skin to induce the immune reaction. No HSV was detected in bullous pemphigoid [43]. Antiviral therapy has a controversial efficacy in HSV-induced erythema multiforme [44,45].

In contrast with HSV-induced EM, mechanisms of Orf-induced autoimmune diseases are still unknown: it has been proven that ORFV infects keratinocytes and dermal fibroblasts, rapidly inducing their death by apoptosis; however, systemic spread of the virus has not yet been demonstrated [46,47]. The latest hypothesis regarding immune-mediated complications includes viral mimicry of host proteins (also known as molecular mimicry) and alteration of basement membrane proteins by the virus (increasing immunogenicity) [20]. Moreover, anti-laminin 332 antibodies were recently suggested to have a potential pathogenetic role in Orf-induced autoimmune blistering disease [25]. In our review, antilaminin 332 circulating autoantibodies were searched for and found to be positive in a single case of immunobullous disease. In the same patient, two months after being treated with oral prednisone, dapsone, and topical clobetasol propionate, no circulating autoantibodies were detectable, perhaps reinforcing the suggested correlation.

The complex interaction among ORFV and the immune system might also be understood through the analysis of ORFV genome organization. ORFV has a linear double-stranded DNA genome containing up to 132 genes organized in a core region and variable regions [5]. While several genetic sequences of the core are quite conserved across most members of the poxvirus group, variable regions at the terminal ends contain genes potentially involved in host range, immune evasion, and immune modulation [5]. The immune response to ORFV involves both innate and adaptive immune reactions. Cell-mediated immunity, and particularly CD4 Th1 cells, play a major role in both primary and reinfection: during the early stage of infection, ORFV induces a vigorous inflammatory response, leading to tissue damage and consequent viral clearance [5]. However, repeated infections can occur in a previously exposed host, despite a prominent inflammatory response, even if clinical manifestations are usually milder in the following episodes [4]. Despite the cause of reinfections still remaining unclear, it may be attributed either to the absence of specific neutralizing antibodies, or to the presence of multiple immunomodulatory factors expressed by the virus, responsible for regulating the host innate and proinflammatory responses [4]. These proteins are mainly encoded by genes in variable regions, and include: viral interleukin 10 orthologue (IL-10, encoded by ORFV127), ORFV interferon resistance protein (IFNr, encoded by ORFV020), chemokine binding protein (CBP, encoded by ORFV112), granulocyte-macrophage colony stimulating factor inhibitory factor (GIF, encoded by ORFV117), NFκB inhibitory protein (ORFV002, ORFV024, ORFV121), deoxy uridine pyrophosphoric acid pyrophosphatase (dUTPase), and BCL2-like apoptosis suppressor (ORFV125) [4,5]. Lastly, ORFV expresses vascular endothelial growth factor (VEGF encoded by ORFV132), which could be responsible for proliferative giant lesions reported in patients with immune deficiencies [48]. These lesions present as large, highly vascularized, tumor-like lesions, and may be mistaken for skin cancer and treated by surgical excision [49].

Regarding Orf diagnosis in humans, according to our systematic review, it was mostly made on the basis of clinical and epidemiological history, while biopsy was performed only in a small percentage of cases [17]. However, considering the wide spectrum of immunological reactions, as well as the poor experience of many clinicians, histopathological examination and direct immunofluorescence may help in achieving a prompt diagnosis.

Speaking about the therapeutic approach, despite the self-limiting nature of Orf, several drugs have been reported for Orf treatment in the literature: imiquimod, cidofovir, interferon, acyclovir, and valacyclovir [37,50,51,52,53]. While adefovir, cidofovir, and valacyclovir are inhibitors of DNA polymerase, imiquimod is an agonist of Toll-like receptors 7 and 8 that induces the release of cytokines, chemokines, and other proinflammatory mediators, stimulating a Th1 cellular immune response [54]. Lastly, interferon has a direct antiviral activity, and decreases viral VEGF activity in vascular tumors [55]. In our review, about a quarter of patients received an initial treatment for primary lesions, mainly consisting of topical or systemic antibiotics, underling the frequent misdiagnosis and consequent overtreatment of this viral self-limiting disease. Treatments for immunological reactions were provided in a higher percentage of cases, suggesting a major clinical impact of these systemic manifestations compared to the localized primary lesions. Based on the immune-mediated nature of complications, topical or systemic corticosteroids were mainly used [16]. A general improvement was reported for all of the patients, regardless of having been treated or not, confirming the self-limiting nature of both Orf lesions and secondary immunological reactions. After the application of imiquimod only on primary lesions, all of them showed a rapid clearance of both Orf and immune-mediated reactions; further research is needed to investigate the potential efficacy of imiquimod [37].

The findings of this systematic review should be seen in the light of some limitations. First, our review is substantially based on case reports and case series, with only one nonrandomized retrospective study. Moreover, the included papers accounted for a small number of patients and this, added to the heterogeneity of the cases’ presentations, necessarily restricted our review to a descriptive analysis. On the other hand, as most of the scientific output on the subject was reported by dermatologists, the main highlight of our work is the presentation of complicated Orf from an infectious diseases specialist’s point of view. This systematic review, written from an infectious disease specialist’s point of view, might also raise awareness in regard to possible ORFV future applications. The ability of ORFV to evade the immune system and to induce immune modulation in the host is an interesting subject of research, raising the hypothesis of using attenuated or recombinant ORFV as a promising new therapeutic agent. Recombinant ORFV has been used as a vector for vaccine delivery for some viral infection in animals, with potential clinical applications in humans [56,57,58]. The short-lived virus-specific induced immunity, together with the absence of virus-neutralizing antibodies, explain the reasons behind its use for repeated immunizations to boost humoral immune responses against the inserted antigens [59]. Even a *Staphylococcus aureus* TRAP (target of RNAIII-activating protein/signal transduction protein) gene vaccine using a live attenuated ORFV vector has been recently described, with promising results [60]. Lastly, attenuated ORFV has been found to have antitumor properties against some types of cancer, thanks to its ability to directly kill human epithelial cancer cells and to initiate a complex antitumor immune response [61]. Promising results are described in melanoma, breast cancer, lung, and colorectal carcinoma [62,63]. Perhaps thanks to these mechanisms, ORFV has also demonstrated to be effective against HBV, HCV, and herpesvirus in several preclinical models [64,65,66].

## 5. Conclusions

To date, there is no efficient vaccine to prevent human Orf. Because of the extensive differential diagnosis, and the wide spectrum of potential immunological reactions, as well as the poor experience of many physicians, human Orf is still underrecognized and often misdiagnosed. In recent years, the SARS-CoV-2 pandemic had a major impact on health systems, radically changing the approach to infectious diseases globally, and generating a fear of new viral outbreaks in the near future [67]. This concern was reinforced by the recent multicountry outbreak of monkeypox [12]. In such an era, all clinicians should be aware of the clinical presentation and the potential complications of human Orf to make a prompt diagnosis, avoid unnecessary investigations and overtreatment, and prevent unnecessary alarmism. A history of working or contact with animals should always be considered in the routinary clinical practice. Erythema multiforme and bullous pemphigoid are the most frequently reported Orf-induced immunological diseases, and the diagnosis is usually based on clinical and epidemiological features; however, histopathological examination and direct immunofluorescence may help in achieving a prompt diagnosis. Despite Orf usually having a self-limiting nature, several drugs have been reported for Orf treatment in the literature. The Orf-related hypersensitivity reactions are similar to cell-mediated immune reactions seen in other viral infections, but its pathophysiology is still unclear. The complex interaction among ORFV and the immune system is an interesting subject of research, and raises the hypothesis of using attenuated or recombinant ORFV as a promising new therapeutic agent for other infections and tumors.

## Figures and Tables

**Figure 1 microorganisms-11-01138-f001:**
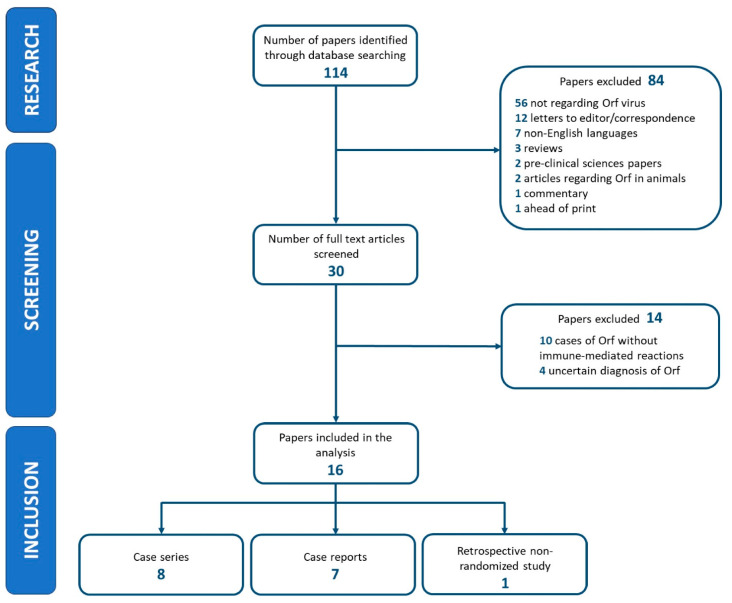
Search strategy and selection process flow-chart.

**Table 1 microorganisms-11-01138-t001:** Summary table regarding articles characteristics, clinical manifestations, diagnosis, and treatment.

	First Author	Ref.	Ps	Animal	Immune-Mediated Reactions	Onset Days	Biopsy	Diagnosis	Autoimmune Complications Treatment
1	Zuelgaray E	[20]	1	Lamb (1)	Epidermolysis bullosa acquisita: sudden widespread pruritic tense blistering eruption and oral mucosal erosions (1)	28 (1)	Yes (1)	Clinical, Bioptic (1)	Topical clobetasol propionate (1)
2	White KP	[24]	2	Sheep (2)	Targetoid palmar lesions, widespread cutaneous and mucosal bullae (1) Widespread cutaneous and mucosal bullae (1)	NA (1) 14 (1)	Yes (2)	Clinical, Bioptic, Molecular (1) Clinical, Bioptic (1)	Corticosteroids (1) Prednisone followed by methotrexate (1)
3	Yilmaz K	[25]	1	Lamb (1)	Tense blisters on the trunk and extremities, with oral mucosal erosions (1)	35 (1)	Yes (1)	Clinical, Bioptic, Molecular (1)	Topical clobetasol propionate, prednisone (1)
4	Agger WA	[26]	1	Lamb/ Goat (1)	Erythema multiforme: extensive blotches in a symmetrical distribution on forehead, neck, and feet (1)	23 (1)	No (1)	Clinical (1)	Prednisone, Schamberg’s solution topical (1)
5	Wilkinson JD	[27]	2	Lamb (2)	Widespread papulovesicular pruritic eruption on skin, with oral and conjunctival involvement (2)	21 (2)	No (1)	Clinical (2)	None (2)
6	Murphy JK	[28]	5	Sheep (2) NA (3)	Bullous pemphigoid: widespread tense blistering eruption (5)	21 (2) 14–21 (3)	Yes (5)	Clinical, Bioptic (5)	Topical clobetasol propionate (2) Topical steroids and saline soaks (3)
7	Huminer D	[29]	1	Lamb (1)	Bullous pemphigoid: erythematous macules and tense bullae on face and extremities (1)	42 (1)	Yes (1)	Clinical, Bioptic (1)	Oral prednisone (1)
8	Kahn D	[30]	1	Sheep (1)	Tense bullae with erythematous borders on trunk and upper extremities (1)	50 (1)	Yes (1)	Clinical, Bioptic (1)	Diphenhydramine (1)
9	Alian S	[11]	1	Sheep (1)	Bullous pemphigoid: widespread maculopapular pruritic eruption with vesiculobullous and target-shaped lesions (1)	14 (1)	Yes (1)	Clinical, Bioptic (1)	Prednisolone (1)
10	Johannessen JV	[31]	18	NA (18)	Erythema multiforme (16) Erythema multiforme bollosum (2)	NA (18)	NA (18)	Clinical (18)	NA (18)
11	López-Cedeño A	[32]	1	Sheep/ Goat (1)	Erythema multiforme: multiple target lesions on hands and limbs (1)	10 (1)	No (1)	Clinical (1)	None (1)
12	Bassioukas K	[33]	2	Sheep (1) NA (1)	Erythema multiforme: maculopapular and target lesions on hands and feet (1) Papulovesicular pruritic eruption on hands, forearms, and neck (1)	14 (1) 18 (1)	Yes (1) NA (1)	Clinical, Bioptic (1) Clinical (1)	Methylprednisolone (1) NA (1)
13	Durdu M	[34]	2	NA (2)	Erythema multiforme: target lesions, erythematous papules, and plaques on hands and forearms (2)	7-14 (2)	Yes (2)	Clinical, Bioptic (2)	Topical corticosteroids (2)
14	Biazar T	[35]	2	Sheep (2)	Erythema multiforme: widespread maculopapular rash with target lesions (2)	25 (1) NA (1)	No (2)	Clinical (2)	Low doses of intravenous corticosteroid and antihistamine (2)
15	Shahmoradi Z	[36]	1	Raw meat (1)	Erythema multiforme: target lesions on hands (1)	14 (1)	No (1)	Clinical (1)	Oral cetirizine, topical steroid, mupirocin, wet dressing with antiseptic solution (1)
16	Erbağci Z	[37]	3	Sheep (1) Raw meat (2)	Erythema multiforme: pruritic erythematous target lesions on hands and feet, unresponsive to topical steroid (1) Erythema multiforme: pruritic erythematous target lesions on limbs, unresponsive to topical steroid and oral antihistamines (1) Recurrent severe oedema of the eyelids, unresponsive to oral antihistamines (1)	NA (3)	No (3)	Clinical (3)	Topical imiquimod on the Orf lesion (3)

Ref = reference; Ps = patients; NA = non-available data.

**Table 2 microorganisms-11-01138-t002:** Localization of the Orf immuno-mediated complications.

Number of Patients	44
Skin involvement	44
Generalized (n, %)	3 (6.8%)
Nongeneralized (n, %)	41 (93.2%)
Limbs and extremities (n, %)	18 (40.9%)
Neck, face, or scalp (n, %)	12 (27.3%)
Trunk (n, %)	8 (18.2%)
Groin (n, %)	7 (15.9%)
Axillae (n, %)	6 (13.6%)
Mucosal involvement	6 (13.6%)
Mouth/lips/nostrils (n, %)	4 (66.7%)
Unspecified mucosal site (n, %)	2 (33.3%)

**Table 3 microorganisms-11-01138-t003:** Orf immuno-mediated complications.

Number of Patients	44
Erythema multiforme (n, %)	26 (59.1%)
Bullous pemphigoid (n, %)	2 (4.5%)
Erythema multiforme bollosum (n, %)	7 (15.9%)
Epidermolysis bullosa acquisita (n, %)	1 (2.3%)
Unspecified diagnosis (n, %)	8 (18.2%)

**Table 4 microorganisms-11-01138-t004:** Histopathology of Orf immuno-mediated lesions. IF = immunofluorescence; BMZ = basement membrane zone.

Number of Biopsies Performed	15
Eosinophilic infiltrate (n, %)	7 (46.7%)
Neutrophilic infiltrate (n, %)	6 (40.0%)
Subepidermal blisters (n, %)	6 (40.0%)
Lymphocytic infiltrate (n, %)	4 (26.7%)
Basal vacuolization (n, %)	1 (6.7%)
Dyskeratosis (n, %)	1 (6.7%)
Necrosis (n, %)	1 (6.7%)
Direct IF testing performed	10
Direct IF BMZ C3 (n, %)	10 (100.0%)
Direct IF BMZ IgG (n, %)	5 (50.0%)
Direct IF BMZ IgA (n, %)	1 (10.0%)

## Data Availability

Not applicable.

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
