# Peer review of "Human Orf with Immune-Mediated Reactions: A Systematic Review"

_microorganisms, 2023, doi:10.3390/microorganisms11051138_

Round 1
Reviewer 1 Report
Manuscript ID: microorganisms-2306380, Systematic Review.
Title: Human Orf and its immune-mediated reactions: a systematic review
Overall assessment.
This paper provides a systematic review of Orf virus infection sin humans, but excludes infections that does not induce immune-mediated pathologies. It is thus very narrowly defined and has limited basis for inference, due to an exclusion of papers that simply report Ord virus infections in general. The review is brief without too much speculation, and it could be suggested that the authors put a bit more perspective on the results to guide future research.
Title. Should be “Human Orf virus with immune mediated reactions…”, since other Orf infections were excluded in the data selection.
Abstract. OK – but needs an update in reference to Line 23. It is a bit odd that the conclusion refer to observations (inappropriate treatments) that is not included in the result section above.
Introduction.
Informative, without being too long. I do however miss information on the possible existence of “biotypes” with reference to more recent studies such as: Coradduzza, E., Sanna, D., Scarpa, F., Azzena, I., Fiori, M. S., Scivoli, R., ... & Puggioni, G. (2022). A Deeper Insight into Evolutionary Patterns and Phylogenetic History of ORF Virus through the Whole Genome Sequencing of the First Italian Strains. Viruses, 14(7), 1473.
M&M. Is OK, but I wonder whether is was a good idea to define the search criteria such that it only yielded 114 papers for screening. However, there seems to be few publication on this particular subject.
Results. OK – but it need to be explained that there is amble literature on Orf virus. The meager results comes from the combination of “Orf” and “Immuno” . As example Google Scholar list more that 7000 paper for the term “Orf virus”. “Human Orf” lists 1680 papers and “human Orf” + Immunotherapy lists 88, i.e, it is the combination that reduce the number of papers.
You need to state that 13 of 16 cases were associate with sheep. No other reservoir/source than sheep could be identified.
Discussion.
Pls. include a short discussion on why you only see cases associated with sheep, when the literature suggests that Orf infection equally often originates from goats. Link this to the paragraphs starting in line 285 – and connect it to your earlier discussion of “biotypes”. Pls. keep it as a perspective, based on tentative and weakly supported indications.
Conclusion
Rarity of reports cannot be used to argue that a disease is underreported. You would not do the same with ebola infection – even though this is much rarer. So please avoid this reasoning in the discussion as well, and perhaps just note that this primarily occupational hazard is rare because few people have direct contact with infected animals.
The conclusion also seems a bit uninspiring – were do we go from here?
Specific comments
Line 32 and onwards: There is a substantial number of “-“ inserted in the words: line 32, 52, 61, 65, 70 etc. that needs to be deleted.
Line 39: Rupicapra rupicapra – in italics.
Line 42. Rewrite so you remove the rather obvious statement that symptoms are present on the udder, only in female animals.
Line 50: The “reasons” appear to be “characteristics”.
Line 61. Add a “.” after “al”
Line 62: delete “be able to”.
Line 75 and the paragraph. Surely the anamnesis (rather that the general epidemiological evidence) would be relevant.
Line 172. Please add the number relevant for calculating the percentage (37.5%, 6/16)
Line 179 – and the rest of the paragraph: as in line 172.
Line 180: write 39.3 sd 15.7 – delete the parenthesis.
Line 180: write (range: 6 to 67).
Line 217. Shorten the sentence i.e., delete “Considering the available data”. Start with “Most”.
Line 217. Shorten the sentence i.e., start with “Twenty-two”.
Line 248: I’m not sure your study “clearly” shows anything – the amount of data is too limited.
Line 265. Comma before “which”.
Line 317. Shorten the sentence i.e., start with “Despite”.
Line 349 and the rest of the paragraph. I do not see the point of this in the given context.
Line 385. Indentation is off
Tables: make it clearer how you calculate the percentages.
Reviewer 2 Report
- Generally, the writing and arguments in the paper are technically valid and the text is easy to read.
- The title is relevant and informative.
- The abstract matches the whole text. It indicates the main topics and results.
- The research question is clearly outlined for me.
- The introduction is concise. Please avoid sub-section in the introduction section.
- Please correct the spelling errors: Ec-thyma (32), Con-tact (65), multi-ple (69), vascu-larised (70), frequently (73), granulu-ma, tuberculosis (77), (78), manifesta-tions (81), and un-der (99)…
- Some references were cited several times including reference 3 which was repeated more than 15 times in the whole text.
- The author has described the aim of the study at the end of the introduction.
- I suggest moving Figure 1 to the Methods section.
- The results are well described
- To assure transparency, the authors reported the limitations of the study.
